# Atomic representation and algorithms for polytomous knowledge spaces

**Zhaorong He** *

Department of Mathematics, Dongguan University of Technology, Dongguan, China

* 17zrhe@stu.edu.cn

**Citation:** He Z (2026) Atomic representation and algorithms for polytomous knowledge spaces. PLoS One 21(4): e0346061. https://doi.org/10.1371/journal.pone.0346061

## Abstract

Classical knowledge space theory provides a rigorous framework for cognitive diagnosis, but its dichotomous response model fails to capture the graded nature of knowledge. While recent research has extended KST to polytomous responses through reductionist approaches, their practical adoption is hindered by computational complexity and the lack of construction methods. This paper introduces a novel framework based on polytomous closure spaces with three key contributions. First, we establish the theory of these spaces alongside an atomic decomposition that enables compact state representation. Second, we characterize granularity conditions that ensure complete atomic decompositions and establish the bijective correspondence between knowledge spaces and their atomic bases. Third, we develop algorithms for base extraction and knowledge space generation that leverage the atomic structure to reduce complex state operations to set computations. The theoretical framework maintains mathematical rigor through lattice-theoretic foundations while achieving computational tractability, providing a practical foundation for adaptive assessment of graded knowledge.

## 1. Introduction

Knowledge space theory (KST), introduced by Doignon and Falmagne [1], provides a rigorous mathematical framework for cognitive diagnosis and adaptive learning. It represents an individual's knowledge state as an element within a structured collection of possible states, known as a knowledge structure. With mathematical foundations deeply rooted in order and lattice theory [2,3,4,5], KST has been successfully implemented in intelligent tutoring systems such as ALEKS [6].

The construction of knowledge structures has been approached through various methodologies, including expert querying [7,8] and analytical procedures [9]. The relationship between skills and items has been formalized through skill maps [10,11] and competence-performance approaches [12,13].

A fundamental result in KST is the Birkhoff-type representation theorem [1,2], which establishes a one-to-one correspondence between quasi-orders on the item

**Data availability statement:** All data are in the manuscript.

**Funding:** This work is supported by National Natural Science Foundation of China (No. 11971287).

**Competing interests:** The authors have declared that no competing interests exist.

set and quasi ordinal spaces. This duality enables knowledge structure construction through either empirical state delineation or logical specification of prerequisite relationships, typically via efficient querying algorithms like the QUERY routine [7,14,15]. For adaptive assessment, the Basic Local Independence Model (BLIM) and related stochastic procedures [16–18] facilitate efficient identification of a student's knowledge state.

A significant advancement was the introduction of learning spaces [6], characterized by the axioms of Learning Smoothness [L1] and Learning Consistency [L2], ensuring seamless learning paths between knowledge states. Learning spaces are equivalent to well-graded knowledge spaces and antimatroids [6,19,20], unifying cognitive, metric, and combinatorial perspectives. Well-gradedness guarantees that any two states are connected by a shortest path where each step modifies mastery of only one item—essential for modeling incremental learning [9]. Algorithmic verification of well-gradedness has been addressed by [20].

Despite its strengths, classical KST is fundamentally limited by its dichotomous response model, reducing item mastery to "correct" (1) or "incorrect" (0). This fails to capture partial knowledge and cannot accommodate polytomous, multi-level scoring systems prevalent in modern educational assessments [21,22], such as those based on Bloom's taxonomy [23–25].

To address this limitation, several researchers have extended KST to polytomous settings. Schrepp [26] initiated this line by considering linearly ordered response sets. Stefanutti et al. [22] proposed a direct generalization with complete lattice response sets, though this approach does not always reduce to classical union-closed knowledge spaces in the binary case, creating theoretical discontinuity [21]. In a pivotal contribution, Heller [21] introduced an indirect reductionist approach based on extended representation, treating each item-value pair as a virtual dichotomous item, thereby embedding the polytomous problem within established dichotomous KST framework while ensuring backward compatibility. More recently, Sun et al. [27,28] have explored well-gradedness and skill mapping, often assuming linear orders. Recent work by Wang et al. [29] further underscores nuanced well-gradedness behavior in polytomous cases, highlighting the importance of modular lattice structures for coherent metrics.

Recent advances in fuzzy skill mapping [28,30] and competency modeling [12,13] further emphasize the need for a comprehensive framework accommodating graded knowledge nature. Cognitive foundations for such an approach are supported by Piaget's stage theory [31,32] and Vygotsky's zone of proximal development [33,34], both emphasizing progressive, multi-faceted learning. Our work also draws inspiration from combinatorial optimization [35] and lattice theory [3–5], providing essential mathematical and combinatorial tools [36].

The reductionist paradigm, while theoretically elegant, introduces fundamental practical challenges for scaling polytomous KST to realistic applications. The core issue stems from the representation gap between the inherently polytomous nature of knowledge and its dichotomous encoding. This gap manifests in several critical aspects. Computationally, representing graded mastery through virtual dichotomous

items leads to combinatorial explosion, making reasoning and assessment intractable for sizable item banks [21]. For instance, with $|Q| = 50$ items and $|V_p| = 4$ levels, the extended representation deals with 200 virtual dichotomous items, rendering state space operations intractable. Moreover, the paradigm lacks natural, scalable algorithms for constructing polytomous structures directly from polytomous response data or expert knowledge, as the underlying representation remains fundamentally dichotomous [22]. Additionally, operational efficiency suffers as state identification and adaptive testing are hindered by the constant need to bridge the representational gap during computational procedures [27].

To bridge this representational gap, this paper introduces a comprehensive framework based on polytomous closure spaces. Our main contributions are:

- **Theoretical Foundation:** We construct the framework of polytomous closure spaces and develop a comprehensive atomic decomposition theory, providing a mathematically rigorous foundation for polytomous knowledge structures based on lattice theory [3,4].

- **Structural Analysis:** We characterize granularity conditions that ensure complete atomic decompositions of knowledge states and establish the bijective correspondence between knowledge spaces and their atomic bases.

- **Algorithmic Design:** Leveraging the atomic decomposition, we design algorithms for base extraction and knowledge space generation that reduce complex state operations to set computations, enabling scalable processing of polytomous knowledge structures.

The remainder of this paper is organized as follows: Section 2 reviews necessary preliminaries and establishes the formal groundwork for polytomous knowledge structures. Section 3 develops the theoretical foundation of polytomous closure spaces and presents the atomic decomposition theory. Section 4 designs Algorithms for base extraction and knowledge space generation that leverage the atomic structure. Section 5 concludes with future research directions.

## 2. Preliminaries and notations

**Definition 2.1 ([3]).** Let $(P, \leq)$ be a poset (partially ordered set).

(1) An element $\top \in P$ is called a **top element** (or greatest element) if $x \leq \top$ for all $x \in P$.

(2) An element $\bot \in P$ is called a **bottom element** (or least element) if $\bot \leq x$ for all $x \in P$.

A poset that has both a top element $\top$ and a bottom element $\bot$ is called a **bounded poset**. A bounded poset $(P, \leq)$ is called **nontrivial** if $\top \neq \bot$ (equivalently, if $P$ contains at least two distinct elements). Otherwise, if $\top = \bot$, we say the poset is **trivial** or **degenerate**.

**Definition 2.2 ([3]).** Let $(P, \leq)$ be a poset.

(1) $(P, \leq)$ is called a **join semilattice** if for any $a, b \in P$, the set $\{a, b\}$ has a supremum in $P$, denoted by $a \vee b$.

(2) $(P, \leq)$ is called a **meet semilattice** if for any $a, b \in P$, the set $\{a, b\}$ has an infimum in $P$, denoted by $a \wedge b$.

A poset $(P, \leq)$ is called a **lattice** if it is both a join semilattice and a meet semilattice.

**Definition 2.3 ([3]).** Let $(P, \leq)$ be a poset.

(1) $(P, \leq)$ is called a **complete join semilattice** if every subset of $P$ has a supremum in $P$.

(2) $(P, \leq)$ is called a **complete meet semilattice** if every subset of $P$ has an infimum in $P$.

A poset $(P, \leq)$ is called a **complete lattice** if it is both a complete join semilattice and a complete meet semilattice.

**Remark 2.1.** The following facts hold:

(i)  In a complete join semilattice, the supremum of the empty set $\bigvee \varnothing$ exists and is equal to the bottom element $\bot$. Similarly, in a complete meet semilattice, $\bigwedge \varnothing$ exists and is equal to the top element $\top$.

(ii)  Every finite lattice is a complete lattice. In particular, every finite lattice possesses both a top and a bottom element.

(iii) For a finite poset $(P,\leq)$, the following are equivalent

   (a)  $(P,\leq)$ is a lattice;

   (b)  $(P,\leq)$ is a complete lattice;

   (c)  $(P,\leq)$ is a complete join semilattice with a top element (equivalently, a complete meet semilattice with a bottom element).

**Definition 2.4 ([4]).** Let $(P,\leq)$ be a poset. A mapping $\varphi : P \to P$ is called a **closure operator** on $P$ if for all $x, y \in P$:

(i)   $x \leq \varphi(x)$;

(ii)  If $x \leq y$, then $\varphi(x) \leq \varphi(y)$;

(iii) $\varphi(\varphi(x)) = \varphi(x)$.

The elements $x \in P$ satisfying $\varphi(x) = x$ are called **closed elements**.

**Definition 2.5.** Let $(L,\leq)$ be a complete join semilattice. An element $x \in L$ is called **completely join-irreducible** (or $\bigvee$**-irreducible**) if

(i)   $x$ is not the least element $\bot$ of $L$;

(ii)  for every nonempty subset $S \subseteq L$, if $x = \bigvee S$ then $x \in S$.

The set of all completely join-irreducible elements of $L$ is denoted by $J_{irr}(L)$.

**Remark 2.2.** In a finite lattice, an element is completely join-irreducible iff it covers exactly one element (or is an atom). The classical definition of join-irreducibility (considering only joins of two elements) is equivalent in the finite case but weaker in general.

**Definition 2.6 ([3]).** A bounded lattice $(L,\leq)$ is **complemented** if for every element $x \in L$, there exists an element $\bar{x} \in L$ such that

$$x \vee \bar{x} = \top \quad \text{and} \quad x \wedge \bar{x} = \bot.$$

The element $\bar{x}$ is called a **complement** of $x$.

Note that an element may have multiple distinct complements in a general complemented lattice.

**Definition 2.7 ([3,4]).** A bounded distributive lattice is called a **Boolean algebra** if it is complemented.

Unlike general complemented lattices where an element may have multiple complements, in a Boolean algebra the complement of each element is unique. This uniqueness allows the complement operation to be treated as a unary operator $\neg : B \to B$ satisfying $\neg(\neg x) = x$ and De Morgan's laws.

Let $Q$ be a non-empty set of items and for each $p \in Q$, let $(V_p, \leq_p)$ be a bounded poset. For notational convenience, we write

$$Q \times V := \bigcup_{p \in Q} (\{p\} \times V_p) \tag{1}$$

for the set of all possible "item-value" pairs. In Heller [21] the same set is denoted by $Q \times V_p$ (see p. 12). We prefer the notation $Q \times V$, which avoids the notational ambiguity that could arise from the subscript $p$ and makes the subsequent exposition cleaner.

 

To represent situations where each item $p$ is assigned exactly one value from its response scale $V_p$, we employ the concept of transversal.

**Definition 2.8 ([35]).** Let $\mathcal{A} = \{A_i \subseteq E : i \in I\}$ be a family of subsets of a set $E$. A **transversal** of $\mathcal{A}$ is a subset $T \subseteq E$ for which there exists a bijection $\psi : T \to I$ such that $x \in A_{\psi(x)}$ for every $x \in T$.

Consider the set $Q \times V$ and its family of subsets $\mathcal{A} = \{\{p\} \times V_p : p \in Q\}$. A subset $K \subseteq Q \times V$ is a transversal of $\mathcal{A}$ if and only if for each $p \in Q$, $K$ contains exactly one element of the form $(p, v)$ with $v \in V_p$. Indeed, define $\psi : K \to Q$ by $\psi((p, v)) = p$; then $\psi$ is a bijection satisfying $(p, v) \in \{p\} \times V_p$, as required by Definition 2.8. Let $V^Q$ denote the set of all transversals of $\mathcal{A}$. There is a natural bijection $\Phi : V^Q \to \prod_{p \in Q} V_p$ given by

$$\Phi(K) = (v_p)_{p \in Q} \quad \text{where } (p, v_p) \in K.$$

For $K \in V^Q$ and $p \in Q$, we denote by $K(p)$ the unique element $v \in V_p$ such that $(p, v) \in K$. Equivalently, $K(p)$ is the $p$-th component of $\Phi(K)$. Thus $K$ can be identified with the tuple $\Phi(K) \in \prod_{p \in Q} V_p$, and we will freely alternate between the two representations.

**Notational Convention.** We adopt the following conventions throughout the paper:

(1) For $K \in V^Q$ and $p \in Q$, $K(p)$ denotes the value assigned to item $p$. Equivalently, $K$ can be represented by the indexed family $(K(p))_{p \in Q}$.

(2) When $Q$ is finite and a specific ordering of its elements is chosen, we represent $K$ as the corresponding $|Q|$-tuple. For example:

   (i) If $Q = \{p_1, \ldots, p_n\}$ with the order given by the indices, we write $K = (K(p_1), \ldots, K(p_n))$.

   (ii) If $Q = \{a,b,c,d\}$ ordered alphabetically, we write $K = (K(a), K(b), K(c), K(d))$.

(3) Let **2** denote the two-element totally ordered set $\{0,1\}$ with $0 < 1$. When $(V_p, \leq_p) = \mathbf{2}$ for every $p \in Q$, there is a natural bijection between $\mathbf{2}^Q$ and the power set $\mathcal{P}(Q)$ of $Q$:

$$\Phi : \mathbf{2}^Q \to \mathcal{P}(Q), \quad \Phi(K) = \{p \in Q : K(p) = 1\}.$$

Using this bijection, we often identify an element $K \in \mathbf{2}^Q$ with the corresponding subset $\Phi(K)$, and write

$$K \cong \{p \in Q : K(p) = 1\}.$$

The symbol $\cong$ here indicates identification via the bijection $\Phi$; it provides an alternative, often more concise, way to denote elements of $\mathbf{2}^Q$.

## 3 Atomic representation for polytomous knowledge spaces

This section establishes the mathematical foundation for atomic representation of polytomous knowledge spaces. We prove that every finite polytomous knowledge space admits a finite generating base consisting of its join-irreducible elements, providing the theoretical basis for algorithmic processing. We further identify granularity as the condition ensuring unique atomic decomposition and bijective correspondence between spaces and their bases.

### 3.1 Polytomous knowledge structures and polytomous knowledge spaces

In this section, we extend the basic notions of knowledge space theory to the polytomous setting, where each item may have more than two ordered response categories. Throughout this section, we assume that each response scale $(V_p, \leq_p)$ is

a *nontrivial* bounded poset (i.e., $\top_p \neq \bot_p$). This excludes degenerate items with only one possible response value, which would carry no diagnostic information.

Let $Q$ be a nonempty set of items. For each item $p \in Q$, we denote by $V_p$ the set of possible response values for that item, and by $\leq_p$ a partial order on $V_p$ where larger values correspond to higher mastery levels. Thus, $(V_p, \leq_p)$ is a bounded poset, with top element $\top_p$ representing full mastery and bottom element $\bot_p$ representing no mastery. We work with the set $V^Q$ and the notation $K(p)$ as established in Section 2. On $V^Q$ we consider the item-wise partial order

$$K \sqsubseteq L \quad \Longleftrightarrow \quad \forall p \in Q, \ K(p) \leq_p L(p) \qquad (K, L \in V^Q).$$

Since each $(V_p, \leq_p)$ is bounded, the product order $\sqsubseteq$ makes $(V^Q, \sqsubseteq)$ a bounded poset as well. Its greatest and least elements are given explicitly by

$$K_\top(p) = \top_p \quad \text{and} \quad K_\bot(p) = \bot_p \qquad (\forall p \in Q),$$

where $\top_p$ and $\bot_p$ are respectively the greatest and least elements of $(V_p, \leq_p)$. We call $K_\top$ the **full knowledge state** and $K_\bot$ the **ignorant state**.

**Definition 3.1.** Let $Q$ be a non-empty set, each $(V_p, \leq_p)$ a nontrivial bounded poset, and $\mathcal{K} \subseteq V^Q$ nonempty. The pair $(\mathcal{K}, \sqsubseteq)$ is called a **polytomous knowledge structure** if

(i)   $\mathcal{K}$ contains the states $K_\top$ and $K_\bot$;

(ii)  for every $p \in Q$ and every $v \in V_p$, there exists $K \in \mathcal{K}$ with $K(p) = v$.

Each element of $\mathcal{K}$ is called a **polytomous knowledge state**, or simply a **state**.

**Remark 3.1.** The following points clarify Definition 3.1 and its relationship to existing work.

(1) *Coverage condition.* Condition (ii) requires that every item–value pair $(p, v)$ is realized by some state. In our framework (states as functions $K : Q \to \bigcup_p V_p$ with $K(p) \in V_p$), this is: $\forall p \in Q, \forall v \in V_p, \exists K \in \mathcal{K} : K(p) = v$. In Heller's transversal view [21] (states as transversals of $\{\{p\} \times V_p\}$), this is equivalently $\bigcup \mathcal{K} = Q \times V$ (where $V = \bigcup_p V_p$). The equivalence follows from the natural bijection between functions and transversals.

Stefanutti et al. [22] impose no coverage condition – their definition only requires $\mathcal{K} \subseteq L^Q$ nonempty, with all items sharing the same lattice $L$. Ge [37] strengthens this by requiring $K_\bot, K_\top \in \mathcal{K}$; Ge [38] follows Stefanutti et al.'s original framework. Wang et al. [39] adopts Heller's uniform definition with $\bigcup \mathcal{K} = Q \times V$.

(2) *Comparison with Heller.* Our definition differs from Heller's [21] in two aspects: (a) we require only a bounded poset for each $(V_p, \leq_p)$, whereas Heller assumes a lattice; (b) we allow $Q$ infinite, while Heller assumes finiteness.

(3) *Comparison with Stefanutti et al. and Ge.* Stefanutti et al. [22] define a polytomous structure as $(Q, L, \mathcal{K})$ where $L$ is a complete lattice (uniform scale) and $\mathcal{K} \subseteq L^Q$ nonempty. They do not require $K_\bot, K_\top \in \mathcal{K}$. Ge [38] follows this framework; Ge [37] strengthens it by explicitly requiring $K_\bot, K_\top$. Both Ge [37,38] further investigate granular structures and their correspondences. In contrast, our definition allows item-specific scales $(V_p, \leq_p)$ that need only be bounded posets, not necessarily lattices. This accommodates items with different rating formats (e.g., 3-point vs. 5-point scales), whereas Stefanutti et al. and Ge require a uniform scale.

(4) *The dichotomous case.* When $(V_p, \leq_p) = \mathbf{2}$ for all $p$, we recover the classical setting. Condition (ii) is automatically satisfied. The isomorphism $\Phi : \mathbf{2}^Q \to \mathcal{P}(Q)$ defined by $\Phi(K) = \{p : K(p) = 1\}$ translates between the two representations.

To develop a theory that includes closure under item-wise suprema, we now strengthen the assumption on response scales: each $(V_p, \leq_p)$ is assumed to be a *nontrivial bounded complete join semilattice*; that is, it has a top element $\top_p$, a

bottom element $\perp_p$ with $\top_p \neq \perp_p$, and every subset has a supremum. Under this assumption, for any family $\mathcal{F} \subseteq V^Q$ we can define its *item-wise supremum* as the element $J_{\mathcal{F}} \in V^Q$ corresponding to the tuple

$$\left(\sup_{V_p}\{K(p) \mid K \in \mathcal{F}\}\right)_{p \in Q} \in \prod_{p \in Q} V_p.$$

Equivalently, $J_{\mathcal{F}}$ is the unique element of $V^Q$ satisfying $J_{\mathcal{F}}(p) = \sup_{V_p}\{K(p) \mid K \in \mathcal{F}\}$ for every $p \in Q$.

**Definition 3.2.** A polytomous knowledge structure $(\mathcal{K}, \sqsubseteq)$ on $V^Q$ is called a **polytomous knowledge space** if for every nonempty subset $\mathcal{F} \subseteq \mathcal{K}$, the item-wise supremum $J_{\mathcal{F}}$ belongs to $\mathcal{K}$. In this case, we also say that $(\mathcal{K}, \sqsubseteq)$ is **closed under item-wise suprema**.

**Remark 3.2.** Every polytomous knowledge space $(\mathcal{K}, \sqsubseteq)$ is automatically a bounded complete join semilattice. Its least element is $K_\top$, its greatest element is $K_\top$, and for any $\mathcal{F} \subseteq \mathcal{K}$, the supremum in $\mathcal{K}$ is computed componentwise: $J_{\mathcal{F}}(p) = \sup_{V_p}\{K(p) \mid K \in \mathcal{F}\}$ for each $p \in Q$. This definition directly generalizes the classical notion of a knowledge space to polytomous items.

Not every polytomous knowledge structure is a knowledge space, as illustrated by the following example.

**Example 3.1.** Consider $Q = \{a,b\}$ and $V_a = V_b = \{0, 1, 2, 3\}$ with the order $0 < 1 < 2 < 3$. Define

$$\mathcal{K} = \{(0, 0), (1, 0), (0, 1), (2, 2), (3, 3)\}.$$

One can check that $\mathcal{K}$ is a polytomous knowledge structure: it contains $K_\perp = (0, 0)$ and $K_\top = (3, 3)$, and for each $p \in \{a, b\}$ and $v \in \{0, 1, 2, 3\}$, there exists $K \in \mathcal{K}$ with $K(p)=v$. However, for $\mathcal{F} = \{(1, 0), (0, 1)\}$, the item-wise supremum $(1,1)$ does not belong to $\mathcal{K}$. Hence $\mathcal{K}$ is not closed under item-wise suprema.

### 3.2 Atoms, join-irreducibility and the finite generation theorem

This subsection establishes the core theoretical result that enables our algorithms: every finite polytomous knowledge space has a finite generating base consisting of its completely join-irreducible elements. We begin by recalling the notion of span, which formalizes the idea of generating a space from a set of elements.

**Definition 3.3.** Let $Q$ be a nonempty set and $V_p$ a nontrivial bounded complete join semilattice for each $p \in Q$. For any $\mathcal{G} \subseteq V^Q$, the **span** of $\mathcal{G}$ is defined as:

$$\mathbb{S}(\mathcal{G}) = \left\{\bigsqcup \mathcal{H} \mid \mathcal{H} \subseteq \mathcal{G}\right\}.$$

We say that $\mathcal{G}$ **spans** $\mathbb{S}(\mathcal{G})$

The central concept for our decomposition theory is that of an *atom*—a minimal knowledge state that contains a particular completely join-irreducible value at a particular item.

**Definition 3.4.** Let $(\mathcal{K}, \sqsubseteq)$ be a polytomous knowledge structure on $V^Q$, where $Q$ is nonempty and for each $p \in Q$, $(V_p, \leq_p)$ is a nontrivial bounded complete join semilattice. For any $p \in Q$ and $v_p \in J_{\text{irr}}(V_p)$, if there exists $A \in \mathcal{K}$ such that

(i) $A(p) = v_p$;

(ii) for any $B \in \mathcal{K}$, if $B \sqsubseteq A$ and $B(p) = v_p$, then $B = A$,

then $A$ is called an **atom** of $\mathcal{K}$ at $(p, v_p)$. Element $K \in \mathcal{K}$ is called an atom of $\mathcal{K}$ if there exist $p \in Q$ and $v_p \in J_{\text{irr}}(V_p)$ such that $K$ is an atom of $\mathcal{K}$ at $(p, v_p)$. The set of all atoms of $\mathcal{K}$ is denoted by $\mathcal{A}(\mathcal{K})$ and is called the **atomic base** of $\mathcal{K}$.

An atom represents the most elementary knowledge component containing a specific join-irreducible value at a specific item. To connect atoms with the order-theoretic notion of join-irreducibility, we need the following property of the value lattices.

**Definition 3.5.** A nontrivial bounded complete join semilattice $(L, \leq)$ is called **supremum-generated** if every element $x \in L$ satisfies

$$x = \bigvee \{ j \in J_{\mathrm{irr}}(L) \mid j \leq x \}.$$

**Remark 3.3.** Every finite lattice is supremum-generated. In the infinite case, supremum-generatedness is a nontrivial condition requiring that every element can be reconstructed from its completely join-irreducible components.

**Theorem 3.1.** Let $(\mathcal{K}, \sqsubseteq)$ be a polytomous knowledge space on $V^Q$ and for each $p \in Q$, $(V_p, \leq_p)$ be a supremum-generated bounded complete join semilattice. Then $\mathcal{A}(\mathcal{K}) = J_{\mathrm{irr}}(\mathcal{K})$.

*Proof.* ($\subseteq$) Assume $K$ is an atom of $\mathcal{K}$, i.e., $K \in \mathcal{A}(\mathcal{K})$. Then there exist $q \in Q$ and $l \in J_{\mathrm{irr}}(V_q)$ such that $K$ is an $l$-atom at $q$. Let $\mathcal{F} \subseteq \mathcal{K}$ be such that $\bigsqcup \mathcal{F} = K$. Then $(\bigsqcup \mathcal{F})(q) = K(q) = l$. Since $l$ is join-irreducible in $V_q$, there exists some $F \in \mathcal{F}$ with $F(q) = l$. As $F \sqsubseteq K$ and $K$ is minimal among states containing $l$ at $q$, we must have $F = K$. Hence $K \in \mathcal{F}$, so $K$ is join-irreducible in $\mathcal{K}$, i.e., $K \in J_{\mathrm{irr}}(\mathcal{K})$.

($\supseteq$): Let $K \in J_{\mathrm{irr}}(\mathcal{K})$, i.e., $K$ is join-irreducible in $\mathcal{K}$. Assume, for contradiction, that $K \notin \mathcal{A}(\mathcal{K})$, meaning that $K$ is not an atom. Since $K$ is not an atom, by Definition 3.4, $K$ fails to be an atom at every pair $(q, w)$ with $q \in Q$, $w \in J_{\mathrm{irr}}(V_q)$, and $w \leq_q K(q)$. This implies that for each such $(q, w)$, there exists a state $J_{q,w} \in \mathcal{K}$ satisfying:

(i) $J_{q,w}(q) = w$;

(ii) $J_{q,w} \sqsubseteq K$;

(iii) $J_{q,w} \neq K$ (otherwise $K$ would be minimal for $(q,w)$).

Consider the set

$$\mathcal{J} = \{ J_{q,w} \mid q \in Q,\ w \in J_{\mathrm{irr}}(V_q),\ w \leq_q K(q) \}.$$

We claim that $\bigsqcup \mathcal{J} = K$. First, since each $J_{q,w} \sqsubseteq K$, we have $\bigsqcup \mathcal{J} \sqsubseteq K$. For the reverse inclusion, fix $q \in Q$. Because $(V_q, \leq_q)$ is supremum-generated,

$$K(q) = \bigvee \{ w \in J_{\mathrm{irr}}(V_q) \mid w \leq_q K(q) \}.$$

For each such $w$, we have $J_{q,w}(q) = w$, hence

$$K(q) \leq_q \bigvee \{ J_{q,w}(q) \mid w \in J_{\mathrm{irr}}(V_q),\ w \leq_q K(q) \} \leq_q (\bigsqcup \mathcal{J})(q).$$

Therefore $K \sqsubseteq \bigsqcup \mathcal{J}$. Thus $\bigsqcup \mathcal{J} = K$. But $K \notin \mathcal{J}$ because every $J_{q,w} \neq K$. This contradicts the assumption that $K$ is join-irreducible in $\mathcal{K}$. Hence $K \in \mathcal{A}(\mathcal{K})$. $\square$

**Theorem 3.2.** Let $Q$ be a finite set, each $(V_p, \leq_p)$ a finite lattice, and $(\mathcal{K}, \sqsubseteq)$ a polytomous knowledge space on $V^Q$. Then $\mathcal{K}$ is finite and

$$\mathcal{K} = \mathbb{S}(J_{\mathrm{irr}}(\mathcal{K})).$$

*Proof.* First, since $Q$ is finite and each $V_p$ is finite, the product set $V^Q = \prod_{p \in Q} V_p$ is finite. As $\mathcal{K} \subseteq V^Q$, it follows that $\mathcal{K}$ is finite. The inclusion $\mathbb{S}(J_{\mathrm{irr}}(\mathcal{K})) \subseteq \mathcal{K}$ is immediate because $J_{\mathrm{irr}}(\mathcal{K}) \subseteq \mathcal{K}$ and $\mathcal{K}$ is closed under arbitrary item-wise joins. For the reverse inclusion $\mathcal{K} \subseteq \mathbb{S}(J_{\mathrm{irr}}(\mathcal{K}))$, we proceed by induction on the height of elements in the poset $(\mathcal{K}, \sqsubseteq)$. Define the

height $h(K)$ of a state $K \in \mathcal{K}$ as the length of the longest chain from the bottom element $K_\perp$ to $K$. Because $\mathcal{K}$ is finite, $h(K)$ is a well-defined nonnegative integer for every $K \in \mathcal{K}$.

**Base case** ($h(K) = 0$): Then $K = K_\perp$. By definition of $\mathbb{S}(J_{\mathrm{irr}}(\mathcal{K}))$, the empty join equals $K_\perp$, so $K_\perp \in \mathbb{S}(J_{\mathrm{irr}}(\mathcal{K}))$.

**Inductive step:** Assume that for all $K' \in \mathcal{K}$ with $h(K') < h(K)$, we have $K' \in \mathbb{S}(J_{\mathrm{irr}}(\mathcal{K}))$. We consider two cases:

**Case 1:** $K$ is join-irreducible in $\mathcal{K}$. Then $K \in J_{\mathrm{irr}}(\mathcal{K})$, and trivially $K = \bigsqcup\{K\} \in \mathbb{S}(J_{\mathrm{irr}}(\mathcal{K}))$.

**Case 2:** $K$ is not join-irreducible in $\mathcal{K}$. Since $(\mathcal{K}, \sqsubseteq)$ is a finite join semilattice (Remark 3.2), every element equals the join of all elements strictly below it. Take $\mathcal{F} = \{F \in \mathcal{K} \mid F \sqsubset K\}$; this set is nonempty because $h(K) > 0$ (so $K \neq K_\perp$). Then $K = \bigsqcup \mathcal{F}$ and $K \notin \mathcal{F}$ by construction. Each $F \in \mathcal{F}$ satisfies $F \sqsubset K$, hence $h(F) < h(K)$. By the induction hypothesis, each $F \in \mathcal{F}$ can be expressed as a join of join-irreducible elements: $F = \bigsqcup S_F$ for some $S_F \subseteq J_{\mathrm{irr}}(\mathcal{K})$. Consequently,

$$K = \bigsqcup_{F \in \mathcal{F}} F = \bigsqcup_{F \in \mathcal{F}} \left( \bigsqcup S_F \right) = \bigsqcup \left( \bigcup_{F \in \mathcal{F}} S_F \right),$$

which expresses $K$ as a join of elements from $J_{\mathrm{irr}}(\mathcal{K})$. Hence $K \in \mathbb{S}(J_{\mathrm{irr}}(\mathcal{K}))$.

By induction, every $K \in \mathcal{K}$ belongs to $\mathbb{S}(J_{\mathrm{irr}}(\mathcal{K}))$. Therefore $\mathcal{K} \subseteq \mathbb{S}(J_{\mathrm{irr}}(\mathcal{K}))$. Combining the two inclusions, we obtain $\mathcal{K} = \mathbb{S}(J_{\mathrm{irr}}(\mathcal{K}))$, completing the proof. □

Theorem 3.2 guarantees that *every* finite polytomous knowledge space—regardless of further structural properties—possesses a finite generating set consisting precisely of its join-irreducible elements. Moreover, under the mild assumption that each value lattice $(V_p, \leq_p)$ is supremum-generated, Theorem 3.1 shows that these join-irreducible elements coincide with the atoms of the space. These results hold without requiring any additional structural conditions. Consequently, the algorithms developed in Section 4, which rely solely on the existence of such a finite generating set, are applicable to *all* finite polytomous knowledge spaces.

**Definition 3.6.** A polytomous knowledge structure $(\mathcal{K}, \sqsubseteq)$ on $V^Q$ is called a **simple polytomous knowledge space** if for every nonempty subset $\mathcal{F} \subseteq \mathcal{K}$, the item-wise infimum $I_{\mathcal{F}}$ belongs to $\mathcal{K}$. In this case, we also say that $(\mathcal{K}, \sqsubseteq)$ is **closed under item-wise infima**.

**Example 3.2** (Finite free combination model). Let $Q = \{a, b\}$ with $V_a = V_b = \{0, 1, 2\}$ (ordered $0 < 1 < 2$). Define

$$\mathcal{K} = \{(0, 0), (1, 0), (0, 1), (1, 1), (2, 0), (0, 2), (2, 2)\}.$$

It is straightforward to verify that $(\mathcal{K}, \sqsubseteq)$ satisfies Definition 3.6: $K_\perp = (0, 0)$ and $K_\top = (2, 2)$ belong to $\mathcal{K}$; every pair $(p, v)$ appears in some state; and for any nonempty $\mathcal{F} \subseteq \mathcal{K}$, the item-wise infimum $\bigsqcap \mathcal{F}$ belongs to $\mathcal{K}$ (a simple case analysis). Thus $(\mathcal{K}, \sqsubseteq)$ is a simple polytomous closure space.

**Example 3.3** (Global synchrony model). Let $Q$ be a nonempty set. For any $p \in Q$, let

$$V_p = \{0, 1\} \cup \{\tfrac{1}{2^{n-1}} \mid n \in \mathbb{N}\},$$

ordered as real numbers, where $\mathbb{N} = \{1, 2, \dots\}$. Thus 1 is the greatest and 0 the least element. Define

$$\mathcal{K} = \{K \in V^Q \mid \exists c \in V, \ \forall p \in Q, \ K(p) = c\},$$

the set of all constant functions. The structure $(\mathcal{K}, \sqsubseteq)$ clearly contains $K_\perp$ (constant 0) and $K_\top$ (constant 1), and saturates $Q \times V$ (for any $v \in V$, the constant function with value $v$ provides the required pair). To see that it is closed under item-wise infima, take any nonempty $\mathcal{F} \subseteq \mathcal{K}$. Each $K \in \mathcal{F}$ is constant: $K(p) \equiv c_K$ for some $c_K \in V$. Let $c^* = \inf\{c_K \mid K \in \mathcal{F}\}$ (the infimum in the real order). Since $V$ is a complete chain, $c^* \in V$. Then the constant function $K^*(p) \equiv c^*$ satisfies $K^* = \bigsqcap \mathcal{F} \in \mathcal{K}$.

Hence $(\mathcal{K}, \sqsubseteq)$ is a simple polytomous closure space. This model represents an extreme scenario where a learner's performance is perfectly synchronized across all items, determined by a single global proficiency level.

**Example 3.4** (Constant affine-subspace model). Let $Q$ be nonempty, $A$ a real affine space. For each $p \in Q$, set $V_p = \mathcal{A}(A) \cup \{\varnothing\}$ (affine subspaces ordered by inclusion). Define

$$\mathcal{K} = \{K \in V^Q \mid \exists M \subseteq A \text{ affine subspace}, \forall p, K(p) = M\}.$$

$K_\top(p) \equiv A$ and $K_\bot(p) \equiv \varnothing$ belong to $\mathcal{K}$, and saturation holds because every affine subspace $M$ yields a constant function in $\mathcal{K}$. For closure under item-wise infima, given $\mathcal{F} \subseteq \mathcal{K}$, each $K \in \mathcal{F}$ is constant with value $M_K$. Then $\bigsqcap \mathcal{F}$ is the constant function with value $\bigcap_{K \in \mathcal{F}} M_K$, which is an affine subspace, hence in $\mathcal{K}$.

**Remark 3.4.** The preceding examples exhibit the flexibility of the simple polytomous closure-space framework:

(i)  Example 3.2 illustrates a finite discrete case with mild inter-item constraints.

(ii)  Example 3.3 represents an extreme global synchrony model.

(iii) Example 3.4 showcases closure spaces with geometric value lattices.

While these structures satisfy the definition, they illustrate that simple closure spaces alone are either overly restrictive (perfect synchrony) or lack mechanisms for encoding rich prerequisite relationships between items. The atomic decomposition theory developed in Section 3.3 will provide a more powerful framework for constructing structured knowledge models that capture meaningful dependencies while maintaining computational efficiency.

**Proposition 3.1.** Let $(\mathcal{K}, \sqsubseteq)$ be a simple polytomous closure space. For any $K \in V^Q$ define

$$\mathrm{cl}_\mathcal{K}(K) = \bigsqcap\{L \in \mathcal{K} \mid K \sqsubseteq L\}.$$

Then $\mathrm{cl}_\mathcal{K}$ is a closure operator on $(V^Q, \sqsubseteq)$ and its set of fixed points is exactly $\mathcal{K}$.

*Proof.* We verify the three axioms of a closure operator:

(i) Since $\mathrm{cl}_\mathcal{K}(K)$ is the greatest lower bound of states containing $K$, we have $K \sqsubseteq \mathrm{cl}_\mathcal{K}(K)$.

(ii) If $K \sqsubseteq M$, then

$$\{L \in \mathcal{K} \mid M \sqsubseteq L\} \subseteq \{L \in \mathcal{K} \mid K \sqsubseteq L\}.$$

The infimum of a larger set is smaller (or equal), hence $\mathrm{cl}_\mathcal{K}(K) \sqsubseteq \mathrm{cl}_\mathcal{K}(M)$.

(iii) First, note that the set $\mathcal{S} = \{L \in \mathcal{K} \mid K \sqsubseteq L\}$ is non-empty because $K_\top \in \mathcal{K}$ and $K \sqsubseteq K_\top$. Since $\mathcal{K}$ is a simple polytomous closure space, it is closed under arbitrary non-empty item-wise infima. Therefore $\mathrm{cl}_\mathcal{K}(K) = \bigsqcap \mathcal{S}$ belongs to $\mathcal{K}$. To prove $\mathrm{cl}_\mathcal{K}(\mathrm{cl}_\mathcal{K}(K)) = \mathrm{cl}_\mathcal{K}(K)$, first note that by (i) we already have $\mathrm{cl}_\mathcal{K}(K) \sqsubseteq \mathrm{cl}_\mathcal{K}(\mathrm{cl}_\mathcal{K}(K))$. For the opposite direction, consider $\mathcal{T} = \{L \in \mathcal{K} \mid \mathrm{cl}_\mathcal{K}(K) \sqsubseteq L\}$. Because $\mathrm{cl}_\mathcal{K}(K) \in \mathcal{K}$, it belongs to $\mathcal{T}$; consequently $\mathrm{cl}_\mathcal{K}(\mathrm{cl}_\mathcal{K}(K)) = \bigsqcap \mathcal{T} \sqsubseteq \mathrm{cl}_\mathcal{K}(K)$. Hence $\mathrm{cl}_\mathcal{K}$ is idempotent.

If $K \in \mathcal{K}$, then $K$ is the least element of $\{L \in \mathcal{K} \mid K \sqsubseteq L\}$, so $\mathrm{cl}_\mathcal{K}(K) = K$. Conversely, if $\mathrm{cl}_\mathcal{K}(K) = K$, then $K$ is the infimum of a non-empty family of elements of $\mathcal{K}$. Since $\mathcal{K}$ is closed under non-empty item-wise infima, $K \in \mathcal{K}$. Thus $\mathrm{Fix}(\mathrm{cl}_\mathcal{K}) = \mathcal{K}$. $\square$

The following proposition establishes a duality between knowledge spaces and closure spaces through complementation.

**Proposition 3.2.** Let $(\mathcal{K}, \sqsubseteq)$ be a polytomous knowledge space where each $(V_p, \leq_p)$ is a Boolean algebra. Define the complement state $\overline{K}(p) = \overline{K(p)}$ and the dual structure $\overline{\mathcal{K}} = \{\overline{K} \mid K \in \mathcal{K}\}$. Then $(\overline{\mathcal{K}}, \sqsubseteq)$ is a simple polytomous closure space.

*Proof.* Since each $(V_p, \leq_p)$ is a Boolean algebra, the complement operation $\overline{\cdot}$ is an order-reversing involution.

(i) $\overline{\mathcal{K}}$ contains $K_\top$ and $K_\bot$. Indeed, $\overline{K_\bot}(p) = \overline{\bot_p} = \top_p$, so $\overline{K_\bot} = K_\top \in \overline{\mathcal{K}}$; similarly $\overline{K_\top}(p) = \overline{\top_p} = \bot_p$, so $\overline{K_\top} = K_\bot \in \overline{\mathcal{K}}$.

(ii) For any $p \in Q$ and $v \in V_p$, let $u = \overline{v} \in V_p$. Because $\mathcal{K}$ is a knowledge space, there exists $K \in \mathcal{K}$ with $K(p) = u$. Then $\overline{K} \in \overline{\mathcal{K}}$ and $\overline{K}(p) = \overline{u} = \overline{\overline{v}} = v$.

Let $\{\overline{K_i}\}_{i \in I} \subseteq \overline{\mathcal{K}}$ be non-empty. For each $p \in Q$, the meet in $V_p$ satisfies

$$\bigwedge_{i \in I} \overline{K_i}(p) = \bigwedge_{i \in I} \overline{K_i(p)} = \overline{\bigvee_{i \in I} K_i(p)} \quad \text{(De Morgan's law)}.$$

Define $L \in V^Q$ by $L(p) = \bigvee_{i \in I} K_i(p)$. Since $\mathcal{K}$ is closed under item-wise joins, $L \in \mathcal{K}$, and therefore $\overline{L} \in \overline{\mathcal{K}}$. By construction, $\overline{L}$ equals the item-wise meet of $\{\overline{K_i}\}_{i \in I}$. □

The mathematical framework developed thus far provides the necessary foundation for the atomic decomposition theory introduced in the next subsection. This theory reveals how every knowledge state can be uniquely represented as a combination of fundamental, indivisible components—a decomposition essential for achieving the computational efficiency required by practical assessment systems.

### 3.3 Granular polytomous knowledge spaces

While Theorem 3.2 guarantees a generating base for every finite space, an additional structural condition—granularity—yields stronger representation properties.

**Definition 3.7.** Let $(\mathcal{K}, \sqsubseteq)$ be a polytomous knowledge structure on $V^Q$. It is said to be **granular** if for any $K \in \mathcal{K}$, any $p \in Q$, and $v \in V_p \setminus \bot_p$, $K(p) = v$ implies that there exists an atom $B$ at $(p, v)$ such that $B \sqsubseteq K$.

**Remark 3.5.** If a polytomous knowledge structure $(\mathcal{K}, \sqsubseteq)$ is granular, then for every $p \in Q$ we necessarily have $V_p \setminus \{\bot_p\} = J_{\mathrm{irr}}(V_p)$; that is, every non-bottom value in $V_p$ must be join-irreducible. Consequently, value lattices such as the diamond lattice (where the top element 3 satisfies $3 = 1 \vee 2$ and is therefore not join-irreducible) cannot appear in a granular structure. Granularity thus imposes a strong constraint on the admissible value lattices, essentially limiting them to chains or other lattices in which no element can be expressed as a nontrivial join of strictly smaller elements.

**Theorem 3.3.** Let $(\mathcal{K}, \sqsubseteq)$ be a granular polytomous knowledge space on $V^Q$ and for each $p \in Q$, $(V_p, \leq_p)$ be a complete lattice. Then

(1) every polytomous knowledge state $K \in \mathcal{K}$ admits a unique decomposition into atoms, i.e.,

$$K = \bigsqcup \{A \in \mathcal{A}(\mathcal{K}) \mid A \sqsubseteq K\}.$$

(2) $(\mathcal{K}, \sqsubseteq)$ is generated by its atomic base, i.e.,

$$\mathcal{K} = \mathbb{S}(\mathcal{A}(\mathcal{K})).$$

*Proof.* (1) Let $K \in \mathcal{K}$ and define $\mathcal{A}(K) = \{A \in \mathcal{A}(\mathcal{K}) \mid A \sqsubseteq K\}$. Let $J = \bigsqcup \mathcal{A}(K)$. Clearly $J \sqsubseteq K$ since each $A \in \mathcal{A}(K)$ satisfies $A \sqsubseteq K$. Suppose for contradiction that $J \sqsubset K$. Then there exists some $p \in Q$ such that $J(p) < K(p)$. Since $\mathcal{K}$ is granular, there exists an atom $A_0 \in \mathcal{A}(\mathcal{K})$ with $A_0 \sqsubseteq K$ and $A_0(p) = K(p)$. But then $A_0 \in \mathcal{A}(K)$ by definition, so $A_0 \sqsubseteq J$, implying $K(p) = A_0(p) \leq J(p)$, contradiction. Therefore $J = K$, i.e., $K = \bigsqcup \mathcal{A}(K)$.

(2) Denote $\mathcal{A} = \mathcal{A}(\mathcal{K})$ and $\widetilde{\mathcal{K}} = \{\bigsqcup S \mid S \subseteq \mathcal{A}\}$. We prove $\mathcal{K} = \widetilde{\mathcal{K}}$. Since $\mathcal{K}$ is a polytomous knowledge space, it is closed under joins of arbitrary subsets and joins are defined item-wise. For any $S \subseteq \mathcal{A} \subseteq \mathcal{K}$, we have $\bigsqcup S \in \mathcal{K}$. In particular, for $S = \varnothing$, $\bigsqcup \varnothing = K_\bot \in \mathcal{K}$. Hence, $\widetilde{\mathcal{K}} \subseteq \mathcal{K}$. On the other hand, let $K \in \mathcal{K}$ be arbitrary. Define the set

$$S_K = \{A \in \mathcal{A} \mid A \sqsubseteq K\}.$$

We show that $K = \bigsqcup S_K$. Obviously, $\bigsqcup S_K \sqsubseteq K$ because every element of $S_K$ is below $K$. Conversely, we need to show that for all $p \in Q$, $K(p) \leq_p \left(\bigsqcup S_K\right)(p)$. If $K(p) = \bot_p$, the inequality holds trivially. If $K(p) = v \neq \bot_p$, then by the granularity of $\mathcal{K}$, there exists a $v$-atom $A \in \mathcal{A}$ at $p$ such that $A \sqsubseteq K$. Therefore, $A \in S_K$ and $A(p) = v$. Consequently,

$$\left(\bigsqcup S_K\right)(p) = \bigvee\{A'(p) \mid A' \in S_K\} \geq_p A(p) = v = K(p).$$

Thus, $K(p) \leq_p \left(\bigsqcup S_K\right)(p)$. It follows that $K = \bigsqcup S_K \in \widetilde{\mathcal{K}}$. Therefore, $\mathcal{K} = \widetilde{\mathcal{K}} = \{\bigsqcup S \mid S \subseteq \mathcal{A}\}$.  □

**Definition 3.8.** Let $(\mathcal{K}, \sqsubseteq)$ be a granular polytomous knowledge space on $V^Q$ and for each $p \in Q$, $(V_p, \leq_p)$ be a complete lattice. For any $K \in \mathcal{K}$, the representation

$$K = \bigsqcup \mathcal{A}(K)$$

is called the **atomic decomposition** of $K$.

**Theorem 3.4.** Let $(\mathcal{K}, \sqsubseteq)$ be a granular polytomous knowledge space. For any $K, L \in \mathcal{K}$:

(1) $\mathcal{A}(K \sqcup L) = \mathcal{A}(K) \cup \mathcal{A}(L)$,

(2) $K \sqsubseteq L$ if and only if $\mathcal{A}(K) \subseteq \mathcal{A}(L)$;

(3) If $\mathcal{K}$ is also meet-closed with item-wise meets, then $\mathcal{A}(K \sqcap L) = \mathcal{A}(K) \cap \mathcal{A}(L)$.

*Proof.* (1) If $A \in \mathcal{A}(K) \cup \mathcal{A}(L)$, then $A \sqsubseteq K$ or $A \sqsubseteq L$, so $A \sqsubseteq K \sqcup L$, hence $A \in \mathcal{A}(K \sqcup L)$. Conversely, if $A \in \mathcal{A}(K \sqcup L)$, then $A \sqsubseteq K \sqcup L = \bigsqcup(\mathcal{A}(K) \cup \mathcal{A}(L))$. Since $A$ is join-irreducible and $A \sqsubseteq \bigsqcup(\mathcal{A}(K) \cup \mathcal{A}(L))$, there must exist some $B \in \mathcal{A}(K) \cup \mathcal{A}(L)$ such that $A \sqsubseteq B$. But $A$ and $B$ are both atoms, so $A \sqsubseteq B$ implies $A = B$, hence $A \in \mathcal{A}(K) \cup \mathcal{A}(L)$.

(2) If $K \sqsubseteq L$, then any $A \in \mathcal{A}(K)$ satisfies $A \sqsubseteq K \sqsubseteq L$, so $A \in \mathcal{A}(L)$. Conversely, if $\mathcal{A}(K) \subseteq \mathcal{A}(L)$, then $K = \bigsqcup \mathcal{A}(K) \sqsubseteq \bigsqcup \mathcal{A}(L) = L$.

(3) If $A \in \mathcal{A}(K \sqcap L)$, then $A \sqsubseteq K \sqcap L \sqsubseteq K$ and $A \sqsubseteq K \sqcap L \sqsubseteq L$, so $A \in \mathcal{A}(K) \cap \mathcal{A}(L)$. Conversely, if $A \in \mathcal{A}(K) \cap \mathcal{A}(L)$, then $A \sqsubseteq K$ and $A \sqsubseteq L$, so $A \sqsubseteq K \sqcap L$, hence $A \in \mathcal{A}(K \sqcap L)$.  □

**Proposition 3.3.** Let $\mathfrak{K}$ be all granular polytomous knowledge spaces on $V^Q$ and for each $p \in Q$, $(V_p, \leq_p)$ be a complete lattice, and $\mathfrak{A} = \{\mathcal{A}(\mathcal{K}) \mid \mathcal{K} \in \mathfrak{K}\}$ their corresponding atomic bases. Then there exists a bijection $F : \mathfrak{K} \to \mathfrak{A}$.

*Proof.* We construct mutually inverse mappings as follows.

(i) Define $F : \mathfrak{K} \to \mathfrak{A}$ by $F(\mathcal{K}) = \mathcal{A}(\mathcal{K})$;

(ii) Define $G : \mathfrak{A} \to \mathfrak{K}$ by $G(\mathcal{A}) = \mathbb{S}(\mathcal{A}) = \{\bigsqcup S \mid S \subseteq \mathcal{A}\}$.

Let $\mathcal{K} \in \mathfrak{K}$ and $\mathcal{A} = F(\mathcal{K}) = \mathcal{A}(\mathcal{K})$. By Theorem 3.3 (2), we have:

$$\mathcal{K} = \mathbb{S}(\mathcal{A}) = G(\mathcal{A}) = G(F(\mathcal{K})).$$

Hence, $G \circ F = \mathrm{id}_{\mathfrak{K}}$. Let $\mathcal{A} \in \mathfrak{A}$. Then $\mathcal{A} = \mathcal{A}(\mathcal{K}_0)$ for some $\mathcal{K}_0 \in \mathfrak{K}$. Let $\mathcal{K} = G(\mathcal{A}) = \mathbb{S}(\mathcal{A})$. We show that $F(\mathcal{K}) = \mathcal{A}(\mathcal{K}) = \mathcal{A}$. First, take any $B \in \mathcal{A}$. We prove $B$ is join-irreducible in $\mathcal{K}$. Suppose $B = \bigsqcup T$ for some $T \subseteq \mathcal{K}$. Since $\mathcal{K} = \mathbb{S}(\mathcal{A})$, each $K \in T$ can be written as $K = \bigsqcup S_K$ for some $S_K \subseteq \mathcal{A}$. Thus,

$$B = \bigsqcup_{K \in T} \left(\bigsqcup S_K\right) = \bigsqcup \left(\bigcup_{K \in T} S_K\right).$$

Let $S = \bigcup_{K \in T} S_K \subseteq \mathcal{A}$. Then $B = \bigsqcup S$. Since $B$ is an atom (in particular, join-irreducible) in the original space $\mathcal{K}_0$ and $S \subseteq \mathcal{A} \subseteq \mathcal{K}_0$, the join-irreducibility of $B$ in $\mathcal{K}_0$ implies that $B \in S$. Hence there exists some $K \in T$ such that $B \in S_K$, i.e.,

$B \sqsubseteq K$. But since $K \sqsubseteq B$ (as $B = \bigsqcup T$ is an upper bound of $T$), we have $K = B$. Therefore $B \in T$, proving that $B$ is join-irreducible in $\mathcal{K}$. Consequently $B \in \mathcal{A}(\mathcal{K})$. Conversely, take any $A \in \mathcal{A}(\mathcal{K})$. Then $A$ is join-irreducible in $\mathcal{K}$ and $A \in \mathcal{K} = \mathbb{S}(\mathcal{A})$, so $A = \bigsqcup S$ for some $S \subseteq \mathcal{A}$. Since $A$ is join-irreducible in $\mathcal{K}$, there exists $B \in S$ such that $A = B$. Hence, $A = B \in \mathcal{A}$. Therefore, $\mathcal{A}(\mathcal{K}) = \mathcal{A}$, i.e., $F(G(\mathcal{A})) = \mathcal{A}$, and so $F \circ G = \mathrm{id}_{\mathfrak{A}}$. $\qquad\square$

Granular polytomous knowledge spaces enjoy a unique atomic decomposition and a bijective correspondence with their bases (Proposition 3.3). Non-granular polytomous knowledge spaces, while lacking these stronger properties, still possess a finite generating base (Theorem 3.2). The following example illustrates this atomic decomposition in a concrete setting.

**Example 3.5** (A granular space). Let $Q = \{a, b\}$ and $V_a = V_b = \{0, 1, 2\}$ with the order $0 < 1 < 2$. Consider the set

$$\mathcal{A} = \{(1, 1),\ (2, 0),\ (0, 2)\}.$$

Let $\mathcal{K} = \mathbb{S}(\mathcal{A})$ be the set of all joins of subsets of $\mathcal{A}$, including the empty join $K_\perp = (0, 0)$. The states in $\mathcal{K}$ are:

$$
\begin{array}{ll}
K_1 = (0, 0) = K_\perp, & K_5 = (2, 1), \\
K_2 = (1, 1), & K_6 = (1, 2), \\
K_3 = (2, 0), & K_7 = (2, 2) = K_\top. \\
K_4 = (0, 2), &
\end{array}
$$

Thus $|\mathcal{K}| = 7$. The space is closed under item-wise joins by construction, since $\mathcal{K} = \mathbb{S}(\mathcal{A})$. Moreover, every pair $(p, v)$ appears at least once, hence $\mathcal{K}$ is a polytomous knowledge space. The set $\mathcal{A}$ is precisely the set of join-irreducible elements (atoms) of $\mathcal{K}$.

We verify that $(1, 1)$ is join-irreducible; the other two elements are treated similarly. Suppose $(1, 1) = \bigsqcup \mathcal{F}$ for some finite non-empty subset $\mathcal{F} \subseteq \mathcal{K}$. Because 1 is join-irreducible in the chain $V_a$, from $(\bigsqcup \mathcal{F})(a) = 1$ we obtain that there exists some $F_a \in \mathcal{F}$ with $F_a(a) = 1$. Similarly, from $(\bigsqcup \mathcal{F})(b) = 1$ we obtain some $F_b \in \mathcal{F}$ with $F_b(b) = 1$. The only state in $\mathcal{K}$ with both $a = 1$ and $b = 1$ is $K_2 = (1, 1)$. Because $(\bigsqcup \mathcal{F})(a) = 1$ and $(\bigsqcup \mathcal{F})(b) = 1$, the set $\mathcal{F}$ cannot consist solely of $K_1 = (0, 0)$. Thus $K_2 \in \mathcal{F}$, proving $K_2$ is join-irreducible in $\mathcal{K}$. Moreover, $K_2$ is minimal with respect to the property of containing the item-value pairs $(a, 1)$ and $(b, 1)$; any proper lower state would miss at least one of these two pairs. Consequently $K_2$ is an atom. The same reasoning shows that each element of $\mathcal{A}$ is an atom.

Consider the state $K_7 = (2, 2)$. Its *atomic decomposition* (in the sense of Definition 3.8) is:

$$K_7 = \bigsqcup \{A \in \mathcal{A}(\mathcal{K}) \mid A \sqsubseteq K_7\} = K_2 \sqcup K_3 \sqcup K_4.$$

Notice that $K_7$ also equals the join of just $K_3$ and $K_4$:

$$K_3 \sqcup K_4 = (2, 0) \sqcup (0, 2) = (2, 2) = K_7.$$

This illustrates that while every state can be expressed as the join of *all* atoms below it (the canonical atomic decomposition), a *minimal* generating subset may also exist.

**Example 3.6** (A non-granular space). Consider $Q = \{a, b, c\}$, where each $p \in Q$ has a diamond-shaped response value lattice $V_p = \{0, 1, 2, 3\}$ with the order structure:

(i)  $0 < 1 < 3$;

(ii) $0 < 2 < 3$;

(iii) 1 and 2 are incomparable,

as is illustrated in Figure?? (Fig 1).

Note that the diamond lattice *cannot appear in any granular polytomous knowledge space* because its top element $3 = 1 \vee 2$ is not join-irreducible, violating the condition $V_p \setminus \{\perp_p\} = J_{irr}(V_p)$ required by granularity. Consider the set

$$\mathcal{A} = \{(1, 1, 0),\ (2, 0, 2),\ (0, 2, 1),\ (0, 1, 2),\ (1, 0, 1)\}.$$

Let $\mathcal{K} = \mathbb{S}(\mathcal{A})$ be the set of all joins of subsets of $\mathcal{A}$, including the empty join $K_\perp = (0, 0, 0)$. The states in $\mathcal{K}$ can be enumerated by considering all subsets of $\mathcal{A}$. A systematic computation yields the following distinct states (each represented as a triple $(v_a, v_b, v_c)$):

$$
\begin{array}{llllll}
K_1 & = & (0,0,0) = K_\perp, & K_8 & = & (1,3,1), \\
K_2 & = & (1,1,0), & K_9 & = & (1,1,2), \\
K_3 & = & (2,0,2), & K_{10} & = & (1,1,1), \\
K_4 & = & (0,2,1), & K_{11} & = & (2,2,3), \\
K_5 & = & (0,1,2), & K_{12} & = & (2,1,2), \\
K_6 & = & (1,0,1), & K_{13} & = & (3,0,3), \\
K_7 & = & (3,1,2), & K_{14} & = & (0,3,3), \\
\end{array}
\qquad
\begin{array}{lll}
K_{15} & = & (1,2,1), \\
K_{16} & = & (1,3,3), \\
K_{17} & = & (3,2,3), \\
K_{18} & = & (3,1,3), \\
K_{19} & = & (3,3,3) = K_\top.
\end{array}
$$

Thus $|\mathcal{K}| = 19$. The space is closed under item-wise joins by construction, since $\mathcal{K} = \mathbb{S}(\mathcal{A})$. Moreover, every pair $(p,v)$ appears at least once, hence $\mathcal{K}$ is a polytomous knowledge space.

The set $\mathcal{A}$ is precisely the set of join-irreducible elements (atoms) of $\mathcal{K}$. We verify this for $(1,1,0)$; the other four elements are treated similarly. Suppose $(1, 1, 0) = \bigsqcup \mathcal{F}$ for some finite non-empty subset $\mathcal{F} \subseteq \mathcal{K}$. Because 1 is join-irreducible in the diamond lattice $V_a$, from $(\bigsqcup \mathcal{F})(a) = 1$ we obtain that there exists some $F_a \in \mathcal{F}$ with $F_a(a)=1$. Similarly, from $(\bigsqcup \mathcal{F})(b) = 1$ we obtain some $F_b \in \mathcal{F}$ with $F_b(b)=1$. Since $(\bigsqcup \mathcal{F})(c) = 0$, every $F \in \mathcal{F}$ must have $F(c)=0$. The only states in $\mathcal{K}$ with $c=0$ are $K_1=(0,0,0)$ and $K_2=(1,1,0)$. Because $(\bigsqcup \mathcal{F})(a) = 1$, the set $\mathcal{F}$ cannot consist solely of $K_1$. Thus $K_2 \in \mathcal{F}$, proving $K_2$ is join-irreducible in $\mathcal{K}$. Moreover, $K_2$ is minimal with respect to the property of containing both $(a,1)$ and $(b,1)$; any proper lower state would miss at least one of these two item-value pairs. Consequently $K_2$ is an atom. The same reasoning shows that each element of $\mathcal{A}$ is an atom.

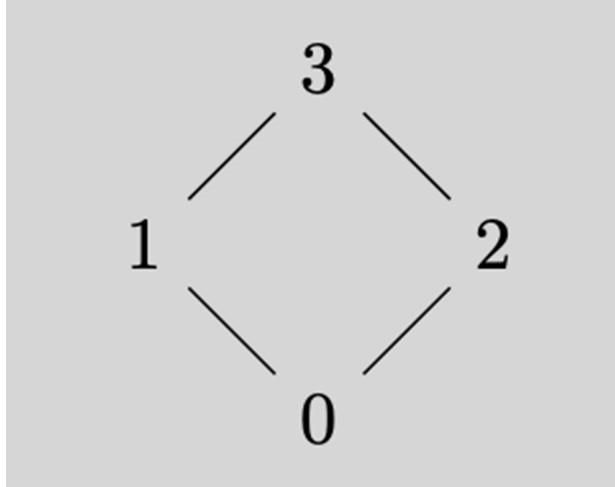

**Fig 1. Diamond-shaped value lattice $V_p$ for each item $p \in Q$.**

Consider the state $K_{11} = (2,2,3)$. Its expression as a join of atoms is:

$$K_{11} = K_3 \sqcup K_4$$

since

$$K_3 = (2, 0, 2), \quad K_4 = (0, 2, 1), \quad K_3 \sqcup K_4 = (2, 0, 2) \sqcup (0, 2, 1) = (2, 2, 3) = K_{11}.$$

Thus $K_{11}$ can be expressed as a join of two atoms, demonstrating that even in a *non-granular* polytomous knowledge space—where the value lattice contains non-join-irreducible elements like 3—states can still be represented as joins of atoms. However, the failure of granularity means that not every non-bottom value $v$ is guaranteed to have an atom supporting it; for example, the value 3 in the diamond lattice lacks such a guarantee, which prevents the space from satisfying Definition 3.7.

The algorithms developed in Section 4 rely *only* on Theorem 3.2; they do *not* require the space to be granular. Hence, they are applicable to both examples above, as well as to any other finite polytomous knowledge space.

## 4 Algorithms for polytomous knowledge spaces

The atomic decomposition established in Section 3 provides a computationally representation of polytomous knowledge structures by transforming state operations into set computations over the base $\mathcal{A}(\mathcal{K})$. This enables three key optimizations:

(1) State ordering $K \sqsubseteq L$ reduces to $\mathcal{A}(K) \subseteq \mathcal{A}(L)$;

(2) Join operations satisfy $\mathcal{A}(K \sqcup L) = \mathcal{A}(K) \cup \mathcal{A}(L)$;

(3) Meet operations (when defined) satisfy $\mathcal{A}(K \sqcap L) = \mathcal{A}(K) \cap \mathcal{A}(L)$.

When $|\mathcal{A}(\mathcal{K})| \ll |Q| \times \max_{p \in Q} |V_p|$, these transformations enable substantial efficiency gains. This section develops practical algorithms for base extraction and knowledge space generation that leverage this atomic structure.

### 4.1 Atomic base extraction for polytomous knowledge spaces

We begin with a straightforward algorithm for identifying join-irreducible elements.

**Algorithm 1 Direct Base Extraction by Definition**

```
Require: Finite knowledge space K = {K₁, ..., K_N}
Ensure: Set of join-irreducible elements A ⊆ K
 1: A ← ∅
 2: for each K ∈ K \ {K⊥} do
 3:   isJI ← True
 4:   properLower ← {L ∈ K | L ⊏ K}
 5:   if properLower = ∅ then
 6:     A ← A ∪ {K}                          ▷ K is an atom
 7:     continue
 8:   end if
 9:   for each nonempty subset S ⊂ properLower do    ▷ Examine all proper subsets
10:     if ⊔S = K then
11:       isJI ← False
12:       break
13:     end if
14:   end for
```

```
15:    if isJI then
16:       𝒜 ← 𝒜 ∪ {K}
17:    end if
18: end for
19: return 𝒜
```

We now analyze the complexity of Algorithm 1. Let $N = |\mathcal{K}|$ be the number of states and $m = |Q|$ the number of items. Each state is represented as an $m$-tuple of values, requiring $O(N \cdot m)$ storage space. The worst-case time complexity is $O(N \cdot 2^N \cdot m)$. For a given state $K$, if we let $p_K = |\{L \in \mathcal{K} \mid L \sqsubset K\}|$ denote the number of states strictly below $K$, the per-state complexity is $O(2^{p_K} \cdot p_K \cdot m)$. The exponential factor $2^N$ makes this algorithm impractical for moderate-sized knowledge spaces: with $N = 20$ items, each state may require checking up to $2^{20} \approx 10^6$ subsets, rendering real-time computation infeasible.

**Example 4.1.** Consider the polytomous knowledge space $(\mathcal{K}, \sqsubseteq)$ from Example 3.6. Applying Algorithm 1: States $K_2 = (1,1,0)$, $K_3 = (2,0,2)$, $K_4 = (0,2,1)$, $K_5 = (0,1,2)$, and $K_6 = (1,0,1)$ each have only $K_1 = (0,0,0)$ strictly below them. Since $\bigsqcup\{K_1\} = K_1 \neq K_i$ for each $i = 2, \ldots, 6$, all five are join-irreducible and are added to the atomic base. For non-join-irreducible states, we find generating subsets: $K_7 = (3,1,2) = \bigsqcup\{K_2, K_3\}$, $K_{11} = (2,2,3) = \bigsqcup\{K_3, K_4\}$, and $K_{19} = (3,3,3) = \bigsqcup\{K_2, K_3, K_4, K_5, K_6\}$ (among many other representations). Hence these states are skipped. The algorithm returns the atomic base

$$\mathcal{A} = \{(1,1,0), (2,0,2), (0,2,1), (0,1,2), (1,0,1)\},$$

which matches the atomic base given in Example 3.6. all five are join-irreducible and are added to the atomic base. (States $K_8$ through $K_{18}$ are similarly found to be non-join-irreducible.)

The direct approach (Algorithm 1) follows exactly the definition of join-irreducibility by enumerating all proper subsets of lower states, but its worst-case time complexity is exponential in $N$ ($O(N \cdot 2^N \cdot |Q|)$). While theoretically correct and useful for illustrating the concept, it becomes impractical for knowledge spaces with more than a dozen states. We now present an alternative algorithm (Algorithm 2) that achieves quadratic time complexity $O(N^2 \cdot |Q|)$ in the worst case by exploiting the structural properties of knowledge spaces. Algorithm 2 is recommended for practical applications with larger state sets, whereas Algorithm 1 serves primarily as a pedagogical tool or for very small spaces.

**Algorithm 2 Base Construction for Polytomous Knowledge Spaces**

```
Require: A polytomous knowledge space (𝒦, ⊑) where
    (i)  𝒦 = {K₁, ..., K_N} is finite and closed under item-wise joins;
    (ii) Q = {p₁, ..., p_m} is a finite item set;
    (iii) Each (V_{p_i}, ≤_{p_i}) is a finite lattice.
Ensure. The set of join-irreducible elements 𝒜 ⊆ 𝒦
1:    Step 1: Sorting
2:    Compute a linear extension of (𝒦, ⊑) such that:
3:      K_i ⊑ K_j ⟹ i < j for all i, j
4:                              ▷ Sorting requires O(N²·m) comparisons
5:    Let K₁ = K_⊥ be the first element in the sorted order
6:    Step 2: Initialization
7:    𝒜 ← ∅                     ▷ Set of discovered join-irreducible elements
8:    Initialize array J[1..N] where each J[i] ∈ 𝒦
9:    for i=1 to N do
10:       J[i] ← K_⊥            ▷ J[i] will accumulate join of join-irreducibles ⊑ K_i
11:    end for
12:   Step 3: Join-irreducible Identification
13:   for i=2 to N do           ▷ Process in sorted order, skip K_⊥
```

```
14:      if J[i] = K_i then
15:        continue            ▷ K_i is generated by previously found join-irreducibles
16:      end if
17:      A ← A ∪ {K_i}          ▷ K_i is join-irreducible
18:      for j = i+1 to N do
19:        if K_i ⊑ K_j then
20:          J[j] ← J[j] ⊔ K_i   ▷ Accumulate join for supersets
21:        end if
22:      end for
23:    end for
24:    return A
```

We now analyze the complexity of Algorithm 2. Let $N = |\mathcal{K}|$ be the number of states and $m = |Q|$ the number of items. Each state is represented as an $m$-tuple of values, requiring $O(N \cdot m)$ storage space for the knowledge states and the $J$ array. The algorithm proceeds in two main phases. First, the states are sorted according to a linear extension of $\sqsubseteq$, which requires $O(N^2 \cdot m)$ time using pairwise comparisons. Second, the join-irreducible identification phase iterates through all states and, for each state $K_i$, updates the $J$ array for all supersets $K_j$ with $j > i$. This involves at most $O(N^2)$ iterations, each requiring $O(m)$ time for join operations and comparisons, yielding a total of $O(N^2 \cdot m)$ for this phase. Thus, the overall worst-case time complexity is $O(N^2 \cdot m)$, which is optimal for pairwise-comparison-based approaches. The space complexity is $O(N \cdot m)$, dominated by storing the knowledge states and the $J$ array.

**Proposition 4.1.** Let $\mathcal{K} = \{K_1, \ldots, K_N\}$ be a finite polytomous knowledge space closed under item-wise joins, and let $K_1, \ldots, K_N$ be a linear extension of $(\mathcal{K}, \sqsubseteq)$ with $K_1 = K_\perp$. Algorithm 2 returns exactly the set $\mathcal{A}(\mathcal{K})$ of join-irreducible elements of $\mathcal{K}$.

*Proof.* We show that after the main loop (Step 3) finishes, the output set $\mathcal{A}$ coincides with $\mathcal{A}(\mathcal{K})$. The argument relies on two invariants maintained throughout the execution. First, at the start of iteration $i$ (with $2 \leq i \leq N$), for every $j \in \{1, \ldots, N\}$ we have

$$J[j] = \bigsqcup \{A \in \mathcal{A} \mid A \sqsubseteq K_j\},$$

so $J[j]$ is the join of all join-irreducible elements discovered so far that lie below $K_j$. Second, all join-irreducible elements among $\{K_1, \ldots, K_{i-1}\}$ are already contained in $\mathcal{A}$.

**Base case** Before iteration $i = 2$ we have $\mathcal{A} = \varnothing$ and $J[j] = K_\perp$ for all $j$; both invariants hold trivially.

**Inductive step** Assume the invariants hold at the start of iteration $i$. We consider two possibilities.

(i)  If $J[i] = K_i$, then by the first invariant $K_i$ equals the join of all join-irreducibles discovered so far that are below $K_i$. Because the order is a linear extension, every element strictly below $K_i$ has index smaller than $i$, and by the second invariant all join-irreducibles among them are already in $\mathcal{A}$. Hence $K_i$ can be expressed as a join of join-irreducible elements strictly below it, which means $K_i$ is not join-irreducible. The algorithm therefore correctly skips $K_i$, and the invariants remain true.

(ii)  If $J[i] \neq K_i$, suppose for contradiction that $K_i$ were not join-irreducible. Then there exists a non-empty subset $S \subseteq \mathcal{K}$ with $\bigsqcup S = K_i$ and $K_i \notin S$. Every element of $S$ is strictly below $K_i$, hence has index $< i$. By the second invariant, all join-irreducible elements occurring in $S$ are contained in $\mathcal{A}$. Because joins are defined item-wise and $\mathcal{K}$ is join-closed, the join of $S$ coincides with the join of those join-irreducibles, which by the first invariant is contained in $J[i]$. Consequently $J[i] = K_i$, contradicting $J[i] \neq K_i$. Thus $K_i$ must be join-irreducible. The algorithm adds $K_i$ to $\mathcal{A}$ and, in the inner loop, updates $J[j] \leftarrow J[j] \sqcup K_i$ for every $j > i$ with $K_i \sqsubseteq K_j$. This update preserves the first invariant because $K_i$ is now a discovered join-irreducible below those $K_j$; the second invariant is clearly maintained as well.

After the last iteration ($i = N$), the second invariant guarantees that every join-irreducible element of $\mathcal{K}$ has been placed in $\mathcal{A}$, while the first case ensures that no non-join-irreducible element has been added. Hence $\mathcal{A} = \mathcal{A}(\mathcal{K})$.

**Example 4.2.** Consider the polytomous knowledge space $(\mathcal{K}, \sqsubseteq)$ from Example 3.6. We apply Algorithm 2 to extract the atomic base $\mathcal{A}$.

**Step 1:** Sorting. One linear extension of $\sqsubseteq$ orders the 19 states as $K_1 = (0, 0, 0)$, $K_2 = (1, 1, 0)$, $K_3 = (2, 0, 2)$, $K_4 = (0, 2, 1)$, $K_5 = (0, 1, 2)$, $K_6 = (1, 0, 1)$, $K_7 = (3, 1, 2)$, $K_8 = (1, 3, 1)$, $K_9 = (1, 1, 2)$, $K_{10} = (1, 1, 1)$, $K_{11} = (2, 2, 3)$, $K_{12} = (2, 1, 2)$, $K_{13} = (3, 0, 3)$, $K_{14} = (0, 3, 3)$, $K_{15} = (1, 2, 1)$, $K_{16} = (1, 3, 3)$, $K_{17} = (3, 2, 3)$, $K_{18} = (3, 1, 3)$, $K_{19} = (3, 3, 3)$.

**Step 2:** Initialization. $\mathcal{A} = \varnothing$; $J[1..19]$ initialized to $K_1 = (0,0,0)$.

**Step 3:** Join-irreducible Identification. For $i = 2, \ldots, 6$, each $K_i$ satisfies $J[i] \neq K_i$, so all five are added to $\mathcal{A}$. Their propagation updates the $J$ array for supersets. Consequently, several later states satisfy $J[i] = K_i$ and are skipped, e.g.,:

(i)   $K_7$: $J[7] = K_7$ (since $K_7 = K_2 \sqcup K_3$);

(ii)  $K_8$: $J[8] = K_8$ ($K_2 \sqcup K_4$);

(iii) $K_9$: $J[9] = K_9$ ($K_2 \sqcup K_5$);

(iv)  $K_{10}$: $J[10] = K_{10}$ ($K_2 \sqcup K_6$);

(v)   $K_{11}$: $J[11] = K_{11}$ ($K_3 \sqcup K_4$);

(vi)  $K_{19}$: $J[19] = K_{19}$ (join of all five atoms).

The algorithm returns $\mathcal{A} = \{(1, 1, 0), (2, 0, 2), (0, 2, 1), (0, 1, 2), (1, 0, 1)\}$, matching the atomic base from Example 3.6.

Example 4.2 demonstrates that in diamond lattices, the intermediate values (1 and 2) are the true atoms, while the top values (3) can be generated as joins of the intermediate atoms, correctly reflecting the lattice structure and thus confirming the algorithm's correctness.

### 4.2 Polytomous knowledge space generation from atomic bases

Once the atomic base $\mathcal{A} = \mathcal{A}(\mathcal{K})$ is identified, Algorithm 3 generates the complete knowledge space by computing joins of all atom subsets via breadth-first search (BFS). The algorithm starts from the ignorant state $K_\perp$ (the empty join) and systematically explores combinations by joining each current state with base atoms. New states are added to both the knowledge space and the exploration queue. The visited set prevents redundancy while ensuring all atom subsets are considered. The BFS approach ensures that each state is generated exactly once, achieving optimal complexity for this generation task. Termination is guaranteed for finite bases and value lattices, with correctness following directly from the atomic decomposition theorem.

### Algorithm 3 Basic BFS Knowledge Space Generation

```
Require: Atomic base A = {A₁, ..., Aₜ} where
    (i) each Aᵢ is a join-irreducible;
    (ii) Q = {p₁, ..., pₘ};
    (iii) each (Vₚ, ≤ₚ) is a finite lattice.
Ensure: Knowledge space K = S(A) generated by the atomic base
1:  K ← {K⊥}                        ▷ Start with empty join (ignorant state)
2:  queue ← [K⊥]                     ▷ BFS queue initialization
3:  visited ← {K⊥}                   ▷ Track visited states
4:  while queue is not empty do
5:    M ← queue.dequeue()
6:    for each A ∈ A do
7:      Kₙₑw ← M ⊔ A                 ▷ item-wise join
```

```
8:        if Kₙₑw ∉ visited then
9:           K ← K ∪ {Kₙₑw}
10:            queue.enqueue(Kₙₑw)
11:            visited ← visited ∪ {Kₙₑw}
12:         end if
13:      end for
14:    end while
15:    return K
```

We now analyze the complexity of Algorithm 3. Let $t = |\mathcal{A}|$ denote the number of atoms (base size), $m = |Q|$ the number of items, and $N = |\mathcal{K}|$ the number of states in the generated space. The algorithm performs a breadth-first search traversal starting from the ignorant state $K_\perp$. Each of the $N$ states is dequeued once, and for each dequeued state, we attempt joins with all $t$ atoms. Each join operation requires $O(m)$ time to compute the componentwise supremum. Thus, the total time complexity is $O(N \cdot t \cdot m)$. The space complexity is $O(N \cdot m)$, dominated by storing the generated knowledge states.

**Example 4.3.** Consider the atomic base $\mathcal{A} = \{(1,1,0),\ (2,0,2),\ (0,2,1),\ (0,1,2),\ (1,0,1)\}$. Applying Algorithm 3:

(i)  Initialization: $\mathcal{K} = \{K_\perp\}$, queue = $[K_\perp]$.

(ii)  First iteration: $M = K_\perp$ generates all five atoms, adding them to $\mathcal{K}$.

(iii)  Subsequent iterations: The BFS explores combinations level by level. For example, with $M=(1,1,0)$ we obtain new states $(3,1,2) = (1,1,0) \sqcup (2,0,2)$, $(1,3,1) = (1,1,0) \sqcup (0,2,1)$, $(1,1,2) = (1,1,0) \sqcup (0,1,2)$, and $(1,1,1) = (1,1,0) \sqcup (1,0,1)$.

After BFS completion, $\mathcal{K}$ contains the 19 states:

$$
\begin{array}{lll}
K_1 = (0,0,0), & K_8 = (1,3,1), & K_{15} = (1,2,1), \\
K_2 = (1,1,0), & K_9 = (1,1,2), & K_{16} = (1,3,3), \\
K_3 = (2,0,2), & K_{10} = (1,1,1), & K_{17} = (3,2,3), \\
K_4 = (0,2,1), & K_{11} = (2,2,3), & K_{18} = (3,1,3), \\
K_5 = (0,1,2), & K_{12} = (2,1,2), & K_{19} = (3,3,3). \\
K_6 = (1,0,1), & K_{13} = (3,0,3), & \\
K_7 = (3,1,2), & K_{14} = (0,3,3), &
\end{array}
$$

This matches Example 3.6, confirming correctness.

**Theorem 4.1.** Let $(\mathcal{K}, \sqsubseteq)$ be a polytomous knowledge space on $V^Q$ with base $\mathcal{A}$. For any $M \in \mathcal{K}$ and $B \in \mathcal{A}$, the following two conditions are equivalent:

(a) For all $K \in \mathcal{K}$, if $M \sqcup B = K \sqcup B$ then $K \sqsubseteq M$;

(b) For all $D \in \mathcal{A}$, if $D \sqsubseteq M \sqcup B$ then $D \sqsubseteq M$.

*Proof.* $(a) \Rightarrow (b)$: Let $D \in \mathcal{A}$ such that $D \sqsubseteq M \sqcup B$. Since $\mathcal{K}$ is a knowledge space, $D \sqcup M \in \mathcal{K}$. We compute

$$(D \sqcup M) \sqcup B = D \sqcup (M \sqcup B) = M \sqcup B$$

where the last equality holds because $D \sqsubseteq M \sqcup B$. By condition $(a)$, we have $D \sqcup M \sqsubseteq M$. But since $M \sqsubseteq D \sqcup M$ by the definition of join, we conclude $D \sqcup M = M$, which implies $D \sqsubseteq M$.

$(b) \Rightarrow (a)$: Let $K \in \mathcal{K}$ such that $K \sqcup B = M \sqcup B$. Since $\mathcal{K}$ is a knowledge space, $K$ can be expressed as the join of base elements: $K = \bigsqcup \{D \in \mathcal{A} \mid D \sqsubseteq K\}$. For any $D \in \mathcal{A}$ with $D \sqsubseteq K$, we have

$$D \sqsubseteq K \sqsubseteq K \sqcup B = M \sqcup B$$

By condition (*b*), this implies $D \sqsubseteq M$. Therefore, all base elements contained in $K$ are also contained in $M$, which means $K \sqsubseteq M$. □

**Corollary 4.1.** For $M \in \mathcal{K}$ and $B \in \mathcal{A}$, the following statements are equivalent:

(a) $M \sqcup B \neq M$;

(b) There exists $D \in \mathcal{A}$ such that $D \sqsubseteq M \sqcup B$ and $D \not\sqsubseteq M$;

(c) $M \sqsubset M \sqcup B$ (i.e., $M \sqsubseteq M \sqcup B$ and $M \neq M \sqcup B$).

*Proof.* The equivalence between (*a*) and (*c*) follows directly from the definition of proper containment. We now prove (*a*) $\Longleftrightarrow$ (*b*). Assume $M \sqcup B \neq M$. By the atomic decomposition, $M \sqcup B = \bigsqcup\{D \in \mathcal{A} \mid D \sqsubseteq M \sqcup B\}$. If all such $D$ satisfied $D \sqsubseteq M$, then we would have $M \sqcup B \sqsubseteq M$, contradicting $M \sqcup B \neq M$. Hence, there exists some $D \in \mathcal{A}$ with $D \sqsubseteq M \sqcup B$ but $D \not\sqsubseteq M$. Conversely, assume there exists $D \in \mathcal{A}$ such that $D \sqsubseteq M \sqcup B$ and $D \not\sqsubseteq M$. If $M \sqcup B = M$, then $D \sqsubseteq M$, contradicting $D \not\sqsubseteq M$. Therefore, $M \sqcup B \neq M$. □

**Algorithm 4 Incremental Construction Illustrating Theorem 4**

```
Require: Base A = {B₁,...,Bₜ} of join-irreducible elements;
    Item set Q = {p₁,...,pₘ};
    Each (Vₚ,≤ₚ) is a finite lattice
Ensure: Knowledge space K = S(A) generated by the base
1:   K ← {K⊥}                    ▷ Initialize with ignorant state
2:   G ← {K⊥}                    ▷ Set of states to be processed
3:   visited ← {K⊥}              ▷ Track visited states
4:   while G ≠ ∅ do
5:     Pick M ∈ G                ▷, e.g., FIFO queue
6:     G ← G \ {M}
7:     for each B ∈ A do
8:       Kₙₑw ← M ⊔ B            ▷ Compute the join
9:       if Kₙₑw = M then
10:         continue ▷ No change; skip
11:       end if                 ▷ By Corollary 2, Kₙₑw ≠ M implies B itself satisfies B ⊑ Kₙₑw
and B ⋢ M.
12:       if Kₙₑw ∉ visited then
13:         K ← K ∪ {Kₙₑw}
14:         G ← G ∪ {Kₙₑw}
15:         visited ← visited ∪ {Kₙₑw}
16:       end if
17:     end for
18:   end while
19:   return K
```

We now analyze the complexity of Algorithm 4. Let $t = |\mathcal{A}|$ denote the number of atoms (base size), $m = |Q|$ the number of items, and $N = |\mathcal{K}|$ the number of states in the generated space. The algorithm processes states from a queue, similar to breadth-first search. Each of the $N$ states is dequeued once, and for each dequeued state, we attempt joins with all $t$ atoms. Each join operation requires $O(m)$ time to compute the componentwise supremum. Thus, the total time complexity is $O(N \cdot t \cdot m)$. The space complexity is $O(N \cdot m)$, dominated by storing the knowledge states and the working sets.

Algorithm 4 demonstrates how Corollary 4.1 guarantees that every non-trivial join $M \sqcup B$ produces a new state (unless it has already been visited). Since the condition of Corollary 4.1 is automatically satisfied by $B$ itself when $B \not\sqsubseteq M$, the

algorithm reduces to the same breadth-first search exploration as Algorithm 3. It is included here primarily to illustrate how the theoretical results of Section 3 inform the design of generation procedures, not as a distinct practical method.

**Example 4.4.** Consider the base

$$\mathcal{A} = \{(1, 1, 0),\ (2, 0, 2),\ (0, 2, 1),\ (0, 1, 2),\ (1, 0, 1)\}$$

from Example 3.6. Applying Algorithm 4 produces the knowledge space $\mathcal{K} = \mathbb{S}(\mathcal{A})$.

(i) Initialization: The queue initially contains only $K_\perp = (0, 0, 0)$.

(ii) First expansion: $M = K_\perp$ is dequeued. For each $B \in \mathcal{A}$, the join $K_{new} = M \sqcup B = B$ differs from $M$, so all five atoms are added to $\mathcal{K}$ and enqueued.

(iii) Subsequent expansions: As states are dequeued, they are joined with every $B \in \mathcal{A}$. For example: When $M = (1,1,0)$ and $B = (2,0,2)$, we obtain $(3,1,2)$, a new state; When $M = (2,0,2)$ and $B = (0,2,1)$, we obtain $(2,2,3)$, another new state.

(iv) Skipping trivial joins: The check $K_{new} = M$ (line 9) prevents redundant processing. For instance, $(1, 1, 0) \sqcup (1, 1, 0)$ yields $K_{new} = M$, so the iteration is skipped.

Corollary 4.1 guarantees that whenever $B \not\sqsubseteq M$, the join $M \sqcup B$ is non-trivial and, unless already present in $\mathcal{K}$, introduces a new knowledge state. After the BFS completes, $\mathcal{K}$ contains the 19 states listed in Example 3.6. The execution demonstrates how Corollary 4.1 guarantees that whenever $B \not\sqsubseteq M$, the join $M \sqcup B$ is different from $M$ (and therefore, unless it has already been generated, represents a new knowledge state).

The two space generation algorithms present complementary pedagogical perspectives. Algorithm 3 employs a straightforward breadth-first search strategy, while Algorithm 4 explicitly illustrates how the theoretical result (Corollary 4.1) guarantees that every non-trivial join $M \sqcup B$ yields a new state. Both algorithms generate the complete knowledge space $\mathcal{K} = \mathbb{S}(\mathcal{A})$, share the same worst-case time complexity $O(|\mathcal{K}| \cdot t \cdot m)$ (where $t = |\mathcal{A}|$), and exhibit identical space complexity $O(|\mathcal{K}| \cdot m)$. Their correctness follows from the finite generation theorem (Theorem 3.2), which ensures that every state in a finite polytomous knowledge space can be expressed as a join of join-irreducible elements. Algorithm 3 is recommended for practical implementations due to its simplicity and clarity. Algorithm 4 is included primarily to illustrate how the theoretical guarantees of Section 3 directly inform the design of generation procedures, even though in this case the theoretical condition does not alter the computational steps.

### 4.3 Algorithm comparison and selection

The four algorithms presented in Sections 4.1 and 4.2 offer complementary approaches to base extraction and space generation. Table 1 summarizes their key characteristics to guide algorithm selection, where $N = |\mathcal{K}|$, $t = |\mathcal{A}|$, $m = |Q|$. Note that space complexity is $O(N \cdot m)$ for all algorithms.

For atomic base extraction,

(i) Algorithm 1 follows directly from the definition of join-irreducibility but has exponential worst-case time complexity, making it suitable only for very small knowledge spaces or for illustrating the concept.

**Table 1. Comparison of algorithms for polytomous knowledge space construction.**

| Algorithm | Type | Time Complexity | Preferred Use Case |
|---|---|---|---|
| Algorithm 1 | Base extraction | $O(N \cdot 2^N \cdot m)$ | Very small spaces, pedagogical |
| Algorithm 2 | Base extraction | $O(N^2 \cdot m)$ | Medium to large spaces |
| Algorithm 3 | Space generation | $O(N \cdot t \cdot m)$ | General purpose, recommended |
| Algorithm 4 | Space generation | $O(N \cdot t \cdot m)$ | Theoretical illustration |

(ii) Algorithm 2 achieves quadratic complexity $O(N^2 \cdot m)$ by exploiting the order structure of the knowledge space and is recommended for practical applications with moderate to large state spaces.

For polytomous knowledge space generation,

(i) Algorithm 3 employs a standard BFS approach, generating all $K_{new} = M \sqcup B$ combinations and filtering via a visited set. Its simplicity and efficiency make it the method of choice for all practical purposes.

(ii) Algorithm 4 includes an explicit comment (line 12) that references Corollary 4.1, highlighting where the theoretical guarantee ensures that each non-trivial join produces a new state.

In practice, Algorithm 2 and Algorithm 3 form the recommended pipeline: extract the base with Algorithm 2, then generate the full knowledge space with Algorithm 3. This combination provides polynomial-time complexity and is scalable to realistic problem sizes. See Examples 4.1 and 4.2 for concrete executions of Algorithms 1 and 2, respectively.

## 5 Conclusion

This paper has presented a comprehensive framework for polytomous knowledge structures that addresses the computational challenges of previous reductionist approaches. Our main contributions are threefold. First, we have established a novel theoretical foundation based on polytomous closure spaces and atomic decomposition. By extending classical knowledge space theory to accommodate graded response patterns through lattice-valued structures, we have provided a mathematically rigorous framework that maintains backward compatibility with dichotomous KST while enabling fine-grained assessment of partial knowledge. Second, we have developed a complete atomic decomposition theory for granular polytomous knowledge spaces. The bijective correspondence between knowledge spaces and their atomic bases enables compact representation of knowledge states, transforming complex state operations into set computations. The granularity condition ensures that every meaningful knowledge component is captured at the atomic level, providing both cognitive interpretability and computational advantages. Third, we have designed practical algorithms that leverage the atomic structure for knowledge space construction and manipulation. The base extraction algorithms (Algorithms 1 and 2) and space generation algorithms (Algorithms 3 and 4) demonstrate how theoretical insights can inform practical implementation, enabling scalable processing of polytomous knowledge structures that was previously hindered by the combinatorial explosion of reductionist encodings. This direct representation avoids the combinatorial explosion inherent in the virtual-item reductionist method, making scalable assessment feasible for realistic item banks with polytomous response options. The framework developed in this work bridges the gap between the inherently polytomous nature of knowledge and computational representation, providing a foundation for real-time adaptive assessment systems capable of capturing partial mastery and learning progression.

Looking forward, several research directions emerge naturally from this work. The atomic decomposition theory could be extended to more general lattice structures, and the relationship between granularity conditions and well-gradedness in polytomous settings presents an important topic for further investigation. Algorithmically, developing incremental update methods for dynamic knowledge structures would enhance practical applicability, though this presents significant challenges within the current atomic framework: when new items are added, expertise is revised, or prerequisite relationships change, the atomic base $\mathcal{A}(\mathcal{K})$ may require non-alocal recomputation, potentially breaking the bijective correspondence between $\mathcal{K}$ and $\mathcal{A}(\mathcal{K})$. Moreover, maintaining granularity under incremental changes is nontrivial. Empirical validation with real educational data and exploration of connections with fuzzy skill mapping approaches offer promising avenues for future work that could further advance the assessment of graded knowledge.

## Acknowledgments

The author would like to express sincere gratitude to the editor and the anonymous reviewers for their valuable feedback and constructive comments, which have greatly improved the quality of this manuscript.

## Author contributions

**Conceptualization:** Zhaorong He.

**Formal analysis:** Zhaorong He.

**Investigation:** Zhaorong He.

**Validation:** Zhaorong He.

**Visualization:** Zhaorong He.

**Writing – original draft:** Zhaorong He.

**Writing – review & editing:** Zhaorong He.

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
