## [Decision Letter · Decision Letter 0]

18 Dec 2025

Dear Dr. He,

Thank you for submitting your manuscript to PLOS ONE. After careful consideration, we feel that it has merit but does not fully meet PLOS ONE’s publication criteria as it currently stands. Therefore, we invite you to submit a revised version of the manuscript that addresses the points raised during the review process.

We look forward to receiving your revised manuscript.

Kind regards,

Fucai Lin, Ph.D.

Academic Editor

PLOS One

3. Please update your submission to use the PLOS LaTeX template. The template and more information on our requirements for LaTeX submissions can be found at http://journals.plos.org/plosone/s/latex

[This work is supported by National Natural Science Foundation of China (No. 11971287).].

Reviewer #1: Accept with Minor Revisions. The manuscript presents significant theoretical and practical contributions to polytomous KST. Addressing the points above will further enhance its clarity, rigor, and readiness for publication. (see the attached file.)

Reviewer #2: The manuscript deals with the efficient atomic representation for polytomous knowledge structures. In my opinion, the topic is interesting and deserves publication in PLOS ONE. On the whole, the manuscript is clear and well-written. Please refer to the attachment for details.

---

## [Author Response · Author response to Decision Letter 1]

3 Mar 2026

Dear Editorial Office,

Please accept our sincere apologies for the delay in submitting this revised manuscript.

All reviewer and editor comments have been addressed. Detailed point-by-point responses to each comment are provided in the attached file "Response-to-Reviewers.pdf".

Due to the mathematical nature of the comments, the responses contain numerous equations and symbols that cannot be properly rendered in this plain text box. The attached PDF provides the complete and correctly formatted responses.

Thank you for your understanding.

Sincerely,

Zhaorong He

---

## [Decision Letter · Decision Letter 1]

15 Mar 2026

Atomic representation and algorithms for polytomous knowledge spaces

PONE-D-25-55989R1

Dear Dr. He,

We’re pleased to inform you that your manuscript has been judged scientifically suitable for publication and will be formally accepted for publication once it meets all outstanding technical requirements.

Kind regards,

Fucai Lin, Ph.D.

Academic Editor

PLOS One

---

## [Editor Report · Acceptance letter]

PONE-D-25-55989R1

PLOS One

Dear Dr. He,

I'm pleased to inform you that your manuscript has been deemed suitable for publication in PLOS One. Congratulations! Your manuscript is now being handed over to our production team.

Kind regards,

on behalf of

Professor Fucai Lin

Academic Editor

PLOS One